# Open-Circuit Fault Diagnosis of Three-Phase PWM Rectifier Using Beetle Antennae Search Algorithm Optimized Deep Belief Network

**Bolun Du**, **Yigang He** * **and Yaru Zhang**

School of Electrical Engineering and Automation, Wuhan University, Wuhan 430072, China;
bolundu93@whu.edu.cn (B.D.); jyldbl@163.com (Y.Z.)
* Correspondence: yghe1221@whu.edu.cn

**Abstract:** Effective open-circuit fault diagnosis for a two-level three-phase pulse-width modulating (PWM) rectifier can reduce the failure rate and prevent unscheduled shutdown. Nevertheless, traditional signal-based feature extraction methods show poor distinguishability for insufficient fault features. Shallow learning diagnosis models are prone to fall into local extremum, slow convergence speed, and overfitting. In this paper, a novel fault diagnosis strategy based on modified ensemble empirical mode decomposition (MEEMD) and the beetle antennae search (BAS) algorithm optimized deep belief network (DBN) is proposed to cope with these problems. Initially, MEEMD is applied to extract useful fault features from each intrinsic mode function (IMF) component. Meanwhile, to remove features with redundancy and interference, fault features are selected by calculating the importance of each feature based on the extremely randomized trees (ERT) algorithm, and the dimension of fault feature vectors is reduced by principal component analysis. Additionally, the DBN stacked with two layers of a restricted Boltzmann machine (RBM) is selected as the classifier, and the BAS algorithm is used as the optimizer to determine the optimal number of units in the hidden layers of the DBN. The proposed method combined with feature extraction, feature selection, optimization, and fault classification algorithms significantly improves the diagnosis accuracy.

**Keywords:** beetle antennae search; deep belief network; modified ensemble empirical mode decomposition; extremely randomized trees; fault diagnosis; three-phase PWM rectifier

## 1. Introduction

Three-phase pulse-width modulating (PWM) rectifiers have been widely used in the fields of electric vehicles, aerospace, renewable energy, high power electrolysis, and military [1]. Compared with the conventional diode or thyristor rectifiers, PWM rectifiers have many merits, e.g., lower harmonic distortion of line current, stabilization, and regulation of the DC-link monitoring signal [2]. However, due to complex operating conditions and unpredictable work performance, the PWM rectifiers are vulnerable to unexpected faults. Once fault occurs, the system runs under abnormal conditions or causes substantial economic losses. Hence, an efficient and accurate fault diagnosis approach is of the utmost to ensure the reliability and security of the PWM rectifiers [3].

In general, the semiconductor switch device faults in power converters are divided into two categories: hard fault (structural fault) and soft fault (parametric fault) [4–6]. Hard faults cause the circuit topology to change due to component damage, resulting in a complete loss of circuit function. The soft fault manifests that the parameter value of the component exceeds the tolerance range of the nominal value. Additionally, the hard faults of the power semiconductor devices are the most common in PWM rectifiers, which can be divided into short-circuit fault (SCF) and open-circuit fault (OCF) [7].

A SCF of the semiconductor switch device will cause an overcurrent, which is very destructive and makes the PWM rectifier shut down immediately. In practice, hardware protection circuits are adopted while a SCF is detected, a fast-acting fuse disconnects for converting a SCF to an OCF [8]. In contrast, an OCF will not immediately cause the shutdown of the system and can remain undetected for an extended period. This may cause overstress on the healthy switches, leading to the second fault of other components [9].

Nowadays, fault diagnosis approaches are classified into model-based approaches [10] and data-driven approaches [11]. Model-based approaches are dependent on the empirical knowledge of the operation conditions, material characteristics, and failure mechanism to build mathematical models, among which the state estimation method, parameter identification method [12], and analytical model method [13] are representative. However, the two-level three-phase PWM rectifier has a symmetrical topology structure with many power semiconductor devices. The mixed-signal, formed by noise and crosstalk of neighboring power semiconductor device, makes the original monitoring signal relatively easy to be distorted, resulting in a low signal-to-noise ratio [14]. Thus, the model-based method can hardly build a precise fault diagnosis model for a two-level, three-phase PWM rectifier. In this case, the data-driven approaches emerge with the advantage that prior expertise on accurate mathematical models is no longer required. Data-driven approaches mainly involve three parts: feature extraction, feature selection, and fault diagnosis.

Currently, numerous feature extraction researches have been widely utilized to capture fault information from the original monitoring signal via time-domain and frequency-domain feature extraction. For time-domain statistical analysis, reference [15] employed kurtosis and entropy of the original monitoring signal as the fault features of the circuit. Long et al. extracted the high-order statistical parameters as features for the diagnosis of the circuit [16]. Nevertheless, these time-domain methods are unable to provide information in specific frequency bands, which makes it challenging to extract useful fault information. For frequency-domain feature extraction, the fast Fourier transform (FFT) is used for spectrum analysis, and the wavelet transform is used for sweep frequency response analysis of the output signal [17]. However, these approaches show poor distinguishability for insufficient fault features for nonlinear and non-stationary signals. Another powerful signal processing method for non-linear and non-stationary signals, named empirical mode decomposition (EMD), ensemble empirical mode decomposition (EEMD) [18], and complete ensemble empirical mode decomposition (CEEMD) [19], has been widely used to solve fault diagnosis of rotating machinery and circuit systems. Additionally, compared with wavelet transform where the basic functions are fixed, the EMD-based method decomposes signals according to time-scale characteristics of data without setting any basis function in advance, which has stronger local stationary. However, the EEMD and CEEMD algorithms are time-consuming, the number of iterations has a great impact on the decomposition effect. Therefore, this paper uses a modified ensemble empirical mode decomposition (MEEMD) [20,21] algorithm to extract fault features of the three-phase PWM rectifier, which not only suppress the mode confusion in the decomposition process, but also reduce the calculation amount.

Feature selection, the most significant step before fault diagnosis, can exclude redundant features and remain representative features [22]. If all the features are imported into the classifier directly without further processing, it will increase the computational complexity. However, there is a common problem concerning what features would make fault diagnosis more accurate. To answer this question, the existing approaches generally applied suitable projections to map the matrices in a feature subspace capturing high-discriminative fault information. A variety of approaches, i.e., independent component analysis (ICA), kernel principal component analysis (KPCA), two-dimensional non-negative matrix factorization (2DNMF), and two directions two-dimensional linear discriminant analysis (TD2DLDA), are implemented to increase the discrimination between different fault categories via further obtaining the lower-dimension feature vectors. Although the above methods allow the user to pick better features and achieve good results for circuit fault diagnosis, there are still drawbacks. For instance, a suitable feature is always difficult to select when the data volume is not large because of insufficient information.

In other words, the features selected in this way are likely not comprehensive, and some useful information may be overlooked. Thus, in this work, the Extremely randomized trees (ERT) algorithm is used to measure the importance of each feature. The best subset of features can be selected via dimensionality reduction.

Nowadays, there are many shallow learning fault diagnosis models, i.e., backpropagation neural network (BPNN), support vector machine (SVM) [23], least squares support vector machine (LSSVM), multiclass relevance vector machine (mRVM) [24], and extreme learning machine (ELM) [25], which have been widely implemented in fault diagnosis. For example, artificial neural network (ANN) is used to implement intelligent classification, in which the dependency and the number of thresholds can be reduced [26]. In [27], an intelligent fault diagnosis method based on an immune neural network is used to acquire fault knowledge of electronic components. Nevertheless, these shallow learning networks can not reveal the complex inherent relationships between the root cause of failure and the signal signatures, which often suffer from invalid learning and weak generalization when learning and training with many fault features. Moreover, various optimization algorithms, such as the genetic algorithm (GA), quantum-behaved, chaos theory, particle swarm optimization (PSO) [16], and crow search algorithm (CAS) [28], have been applied to optimize the hyper-parameters of the above shallow learning models. Hereafter, deep learning models have been emerged as a practical approach due to its powerful generalization ability by learning the mapping relationship between the available fault feature and the corresponding fault category. Currently, several effective deep learning models have been applied in fault diagnosis, i.e., deep belief network (DBN) [29], sparse auto-encoder (SAE). For instance, Sun et al. [28] presented a novel DBN model optimized by the CAS to realize fault diagnosis for a DC-DC circuit. In [30], Wen et al. investigated a new deep transfer learning method for fault classification, which is a supervised transfer learning based on a three-layer SAE. In [31], the proponent of the DBN algorithm said that DBN could overcome the limitation of shallow neural networks. DBN is composed of multi-layer units, which can learn to obtain a feature vector that is more suitable for classification. However, the performance of DBN is very vulnerable to the change of DBN structure, such as the depth of the model and the number of hidden layer units. In [32], extensive experiments had been carried out by Coates et al., and the results showed that the number of hidden layer units had a more critical effect on the performance of DBN than the depth. It is necessary to propose a suitable optimization algorithm to determine the number of hidden layer units of DBN.

Consequently, this paper proposes a novel fault diagnostic approach for a two-level three-phase PWM rectifier based on beetle antennae search optimized deep belief network (BAS-DBN). The main contributions of this paper are summarized as follows:

(1) As an improved EMD-based algorithm, MEEMD overcomes the shortcomings of EEMD and CEEMD. It has less computation time and higher reconstruction accuracy when decomposing the original signal into more representative intrinsic mode function (IMF) components. For fully mining sensitive features, the ERT algorithm is proposed to analyze features from multiple respects to obtain the optimal feature set. Feature selection can avoid feature redundancy and overfitting, thereby improving the accuracy of the fault classifier and constructing a faster and lower-consumption fault diagnosis model.

(2) The DBN can find out the essential structure of the data through the layer-by-layer nonlinear mapping and finally realize the deep extraction of features. The BAS algorithm is used to optimize the number of hidden nodes in DBN, avoiding critical deficiencies such as the premature convergence to sub-optimal solutions. Simulation results show that the proposed method achieves higher accuracy by comparing it with the other shallow learning models and optimization algorithms.

The rest of this paper is organized as follows. Section 2 presents the methodologies and theoretical of feature extraction, feature selection, and fault diagnosis algorithms. In Section 3, the simulation model of a two-level three-phase PWM rectifier is presented, and the fault categories are analyzed. Section 4 presents the experimental results of different classification methods compared with BAS-DBN. The conclusion and future researches are presented in Section 5.

## 2. Proposed Framework & Theoretical

The proposed fault diagnostic strategy for a two-level three-phase PWM rectifier is represented in Figure 1, and the detailed description is illustrated as follows:

Step 1: The healthy condition and fault modes for a two-level three-phase PWM rectifier are defined. The fault monitoring signal and the reference signal under the healthy condition and different OCFs are sampled from the two-level three-phase PWM rectifier.

Step 2: The initial feature vectors are extracted from the monitored current signals based on MEEMD. In detail, time-domain, frequency-domain, and energy characteristics of each IMF component are computed as the circuit fault features.

Step 3: The ERT algorithm calculates the importance of each fault feature, and the threshold value is set to remove the features with redundancy and interference. Afterward, the principal component analysis (PCA) algorithm is used to reduce the dimension of fault feature vectors for decreasing the calculation costs and improving the efficiency of fault diagnosis.

Step 4: The optimized DBN-BAS algorithm is utilized to achieve an intelligent fault diagnosis of the two-level three-phase PWM rectifier by optimizing and determining the optimal number of the neurons in the first and second hidden layers of DBN.

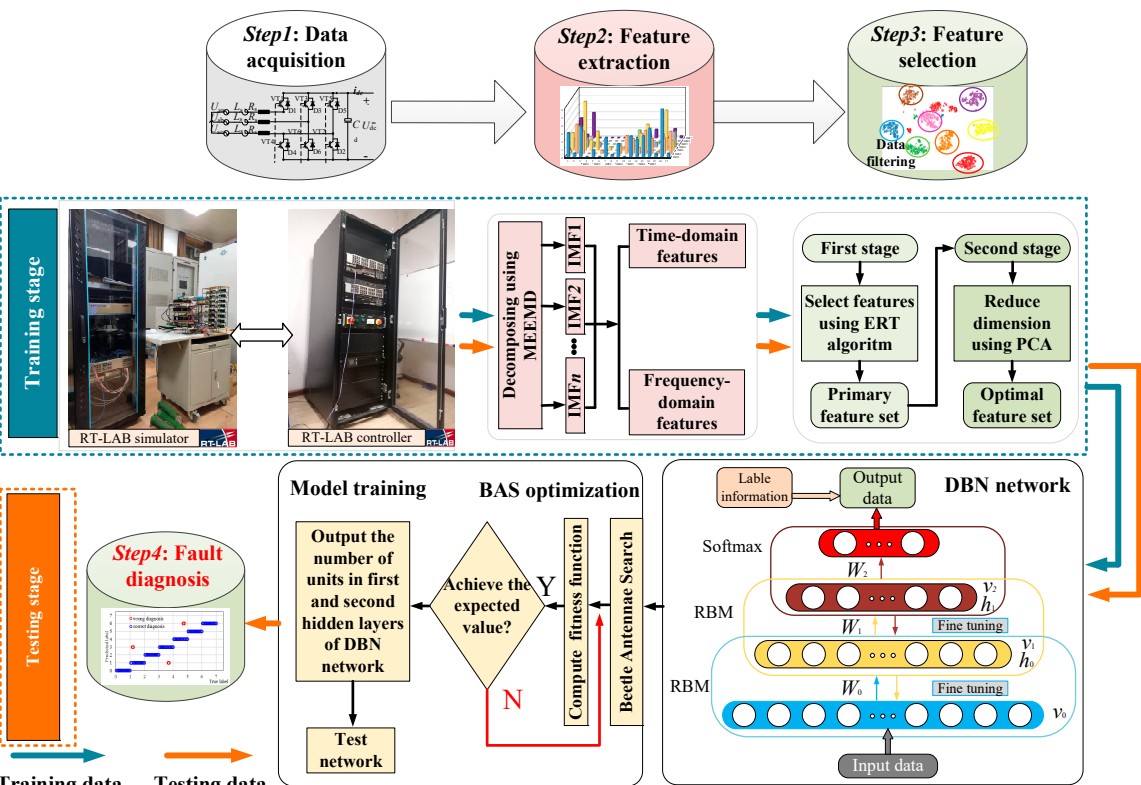

**Figure 1.** The framework of the proposed fault diagnostic approach based on beetle antennae search algorithm optimized deep belief network.

### 2.1. Modified Ensemble Empirical Mode Decomposition

The essence of the MEEMD algorithm [33] is to use a certain rule to separate the abnormal signals in the original data, and then perform EMD decomposition on the remaining signals. Such processing can not only ensure the completeness of the original data, but also reduce the influence of abnormal signals on the decomposition results. The MEEMD algorithm avoids these problems by introducing the permutation entropy (PE) to randomly detect the abnormal signals. The steps of MEEMD are as follows:

Step 1: Add the positive and negative paired white noise $n_i(t)$ and $-n_i(t)$ into the original signal $x(t)$ to obtain a new sequence:

$$\begin{cases} x_t^+(t) = x(t) + a_i n_i(t), t = 1, 2, \ldots, N_e \\ x_t^-(t) = x(t) - a_i n_i(t), t = 1, 2, \ldots, N_e \end{cases} \tag{1}$$

where $a_i$ is the amplitude of the white noise signal. $n_i(t)$ represents the white noise, of which the root mean square value should be close to the root mean square value of $x(t)$. $N_e$ denotes the logarithm of the white noise, generally not higher than 100. Perform an EMD algorithm on $x_t^+(t)$ and $x_t^-(t)$ to obtain the IMF component series $\{l_{i1}^+(t)\}$ and $\{l_{i1}^-(t)\}(i = 1, 2, \ldots, N_e)$, from which the first IMF component $I_1(t)$ can be obtained via ensemble averaging.

$$I_1(t) = \frac{1}{2N} \sum_{i=1}^{Ne} [I_{i1}^+(t) + I_{i1}^-(t)] \tag{2}$$

Step 2: Based on the permutation entropy $\delta$ of the obtained IMF component, if the permutation entropy of the IMF component is greater than the threshold, it is an abnormal component. Otherwise, it is a stationary component. If $I_1(t)$ is an abnormal component, continue to step 1 until the obtained IMF component $I_1(t)$ is no longer abnormal.

Step 3: The abnormal components are separated from the original signal, and then the remaining is decomposed by the EMD algorithm. Finally, arrange all the IMF components obtained from high frequency to low frequency.

$$r(t) = x(t) - x'(t) \tag{3}$$

$$r(t) \overset{EMD}{\rightarrow} \sum_{k=1}^{m} I_k(t) + r(t) \tag{4}$$

where $x'(t)$ represents the sum of all abnormal signals, $r(t)$ denotes the residual signals, and $I_k(t)$ is the $k$th IMF components obtained via the MEEMD algorithm.

## 2.2. Extremely Randomized Trees

The ERT algorithm [34], which is proposed by Pierre Geurts et al., calculates the variable importance measures (VIM) of feature by calculating the purity of decision tree nodes by the Gini index. At last, a certain proportion of features are deleted according to the VIM value to obtain an optimal feature set.

Assuming that there are $m$ features $X_1, X_2, \ldots, X_m$, the VIM value of each feature is expressed as $VIM_j^{(Gini)}$, representing the average change in the impurity purity of the $j$th feature in the ERT decision trees. The formula for calculating the Gini index is as follows:

$$GI_m = \sum_{k=1}^{K} \sum_{k' \neq k} p_{mk} p_{mk'} = 1 - \sum_{k=1}^{K} p_{mk}^2 \tag{5}$$

where $K$ represents the number of categories with samples. $p_{mk}$ represents the proportion of category $k$ in node m, and $p_{mk'} = 1 - p_{mk}$.

For the importance of feature $X_j$ at node $m$, the variation of the Gini index before and after the branch of node $m$ is expressed as follows:

$$VIM_{jm}^{(Gini)} = GI_m - GI_l - GI_r \tag{6}$$

where $GI_l$ and $GI_r$ represent the Gini index of the two new nodes after branching, respectively.

If the node of the feature in the decision tree $i$ is in the set $M$, then the importance of $X_j$ in the $i$th tree is expressed as follows:

$$\begin{cases} VIM_{ij}^{(Gini)} = \sum_{m \in M} VIM_{jm}^{(Gini)} \\ VIM_{j}^{(Gini)} = \sum_{i=1}^{n} VIM_i \end{cases} \tag{7}$$

Ultimately, the importance score of the feature is obtained by normalization as follows:

$$VIM_j = \frac{VIM_j}{\sum_{j=1}^{m} VIM_j} \tag{8}$$

### 2.3. Deep Belief Network

The concept of DBN put forward by Hinton et al. in 2006 was an area of machine learning research, which overcame the limitations of shallow network methods. It is constructed from multiple layers of restricted Boltzmann machines (RBMs), which can extract deep-seated features from complex data. DBN can be viewed as the stacking of simple learning modules. DBN training consists of unsupervised layer-by-layer pre-training and supervised fine-tuning. The former achieves complex nonlinear mapping by directly mapping data from input to output, which is also the critical factor for its robust feature extraction capability. After pre-training, the DBN is trained, supervised by adding a classifier at the top level of DBN to reduce training error. This classifier uses a backpropagation algorithm to fine-tune the relevant parameters of the DBN.

As shown in Figure 2, the schematic representation contains three stacked RBMs. The input layer is the visible layer, which is composed of n visible units $v = (v_1, v_2, \cdots\cdots, v_n)$. Hidden1 is the first hidden layer, which is composed of $m$ hidden units $h = (h_1, h_2, \cdots\cdots, h_m)$. Both are binary random vectors, i.e., $v \in \{0,1\}^n$, $h \in \{0,1\}^m$. Since RBM is an energy-based model, the energy function $E(v, h|\theta)$ is defined as follows:

$$\begin{aligned} E(v, h|\theta) &= -\alpha^T v - \beta^T h - v^T w h \\ &= -\sum_{i=1}^{n} \alpha_i v_i - \sum_{j=1}^{m} \beta_j h_j - \sum_{i=1}^{n} \sum_{j=1}^{m} v_i w_{ij} h_j \end{aligned} \tag{9}$$

where $\theta = [\alpha, \beta, w]$, $\alpha_i$ and $\beta_j$ represent the bias of $v_i$ and $h_j$; $w_{ij}$ is the weight that connects $v_i$ and $h_j$. Then, the probability distribution to every possible pair of $v$ and $h$ can be defined as the following energy function

$$p(v, h) = \frac{1}{Z} \exp(-E(v, h)) \tag{10}$$

where $Z$ is the normalizing constant, as expressed in Formula (11). It can be calculated by summing all possible pairs of $v$ and $h$

$$Z(\theta) = \sum_{v} \sum_{h} \exp(-E(v, h)) \tag{11}$$

The probability that the network assigns to $v$ is as follows:

$$p(v|\theta) = \sum_{h} p(v, h) = \frac{1}{Z} \sum_{h} \exp(-E(v, h)) \tag{12}$$

Furthermore, there is a bidirectional connection between the hidden layer and visible layer, while the neurons in the same layer are independent of each other. When the visible layer is determined, the conditional probability of the visible layer units is presented as follows:

$$\begin{aligned} p(h|v; \theta) &= \frac{p(v, h; \theta)}{p(v; \theta)} = \prod_{j} p(h_j|v) \\ p(v|h; \theta) &= \frac{p(v, h; \theta)}{p(h; \theta)} = \prod_{j} p(v_i|h) \end{aligned} \tag{13}$$

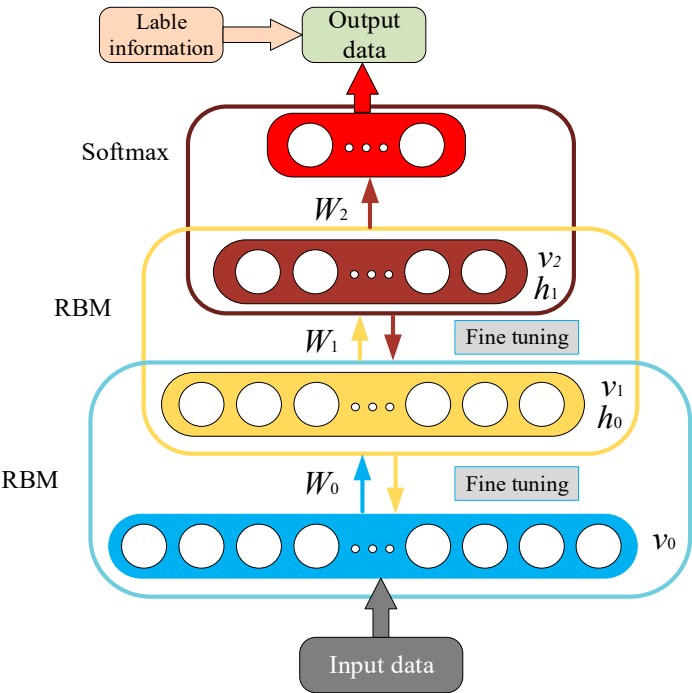

**Figure 2.** DBN with two hidden layers.

The function $sig(x) = 1/(1+e^{-x})$ can be used to calculate the following activation probabilities

$$\begin{cases} p(h_j = 1|v) = sig(\beta_j + \sum_{i=1}^{n} w_{ij}v_i) \\ p(v_i = 1|h) = sig(\alpha_i + \sum_{i=1}^{n} w_{ij}h_j) \end{cases} \tag{14}$$

Given the training data, the probability $p(v)$ of Formula (12) can be maximized by adjusting corresponding parameters. The probability of a training vector is related to the energy of the vector. Therefore, the parameters of RBM can be estimated based on the principle of maximum likelihood estimation. The log-likelihood derivative of $\theta$ can be derived as follows:

$$\frac{\partial \log p(v)}{\partial \theta} = -\sum_{h} p(h|v) \frac{\partial E(v,h)}{\partial \theta} + \sum_{v}\sum_{h} p(v|h) \frac{\partial E(v,h)}{\partial \theta} = -\left\langle \frac{\partial E(v,h)}{\partial \theta} \right\rangle_0 + \left\langle \frac{\partial E(v,h)}{\partial \theta} \right\rangle_\infty \tag{15}$$

where $\left\langle \frac{\partial E(v,h)}{\partial \theta} \right\rangle_0 a$ and $\left\langle \frac{\partial E(v,h)}{\partial \theta} \right\rangle_\infty$ denote the expectation of $p(h|v)$ concerning the data distribution and the model, respectively. However, it is quite challenging to attain an unbiased sample of $\langle \cdot \rangle_{model}$. The learning rule is similar to the objective gradient function named contrastive divergence, where $\langle \cdot \rangle_{model}$ can be replaced by $k$ iterations of Gibbs sampling. Therefore, according to Formula (15), the update rules of the model parameters are as follows:

$$\begin{cases} \Delta w_{ij} = \rho(\langle v_i h_j \rangle_0 - \langle v_i h_j \rangle_k) \\ \Delta \alpha_i = \rho(\langle v_i \rangle_0 - \langle v_i \rangle_k) \\ \Delta \beta_j = \rho(\langle h_j \rangle_0 - \langle h_j \rangle_k) \end{cases} \tag{16}$$

where $\rho \in (0,1)$ is the learning rate.

### 2.4. DBN Optimized by Beetle Antennae Search Algorithm

In this paper, a DBN with two hidden layers is selected for fault diagnosis of a two-level three-phase PWM rectifier. The BAS optimization algorithm is used to the optimal number of neurons in the hidden layer of the DBN. Similar to the GA and PSO optimization algorithms, BAS can automatically implement the optimization process without knowing the specific form of function and gradient information. Furthermore, there is only one individual, and the speed of optimization has been significantly improved. The dimension of the search space in BAS is 2.

The biological principle of the BAS algorithm can be interpreted that the two antennae of the beetle judge the strength of the food odor on the left and right sides to determine the direction in the next step. The flow chart of the BAS algorithm can be summarized in Figure 3, which can be divided into the following steps:

(1) Suppose there is a k-dimensional optimization space, $x_{left}$ and $x_{right}$ represent the coordinates of the left and right antennae of the beetle, respectively. $x^t$ represent the centroid position of the beetle at time $t$, and $d_0$ represent the distance between the two antennae. If the initial orientation of the beetle is random, the vector that the left antennae of the beetle point to the right antennae is also arbitrary. Hence, a normalized random vector is assumed as follow

$$\vec{b} = rands(k,1)/\|rands(k,1)\| \tag{17}$$

$$x_{left} - x_{right} = d_0 \cdot \vec{b} \tag{18}$$

where $x_{left}$ and $x_{right}$ can be expressed as the centroid position

$$\begin{cases} x_{left} = x^t + d_0 \cdot \vec{b}/2 \\ x_{right} = x^t - d_0 \cdot \vec{b}/2 \end{cases} \tag{19}$$

(2) The objective function is set as $f(\cdot)$ and the objective function value at the two position coordinates of the left and right antennae are calculated as $f(x_{left})$ and $f(x_{right})$. Compare the size of these two values and choose the right or left step of the beetle position according to the optimization direction of the objective function $\delta'$.

(3) Subsequently, the beetle's centroid position at time $t + 1$ is updated as follows:

$$x^{t+1} = x^t - \delta' \cdot \vec{b} \cdot sign(f(x_{left}) - f(x_{right})) \tag{20}$$

The fitness function is set as follow

$$MSE = \frac{1}{N} \sum_{i=1}^{N} \|y_{pre} - y_{true}\|^2 \tag{21}$$

where $y_{pre}$ denotes the output value of the DBN classifier and $y_{true}$ denotes the actual value.

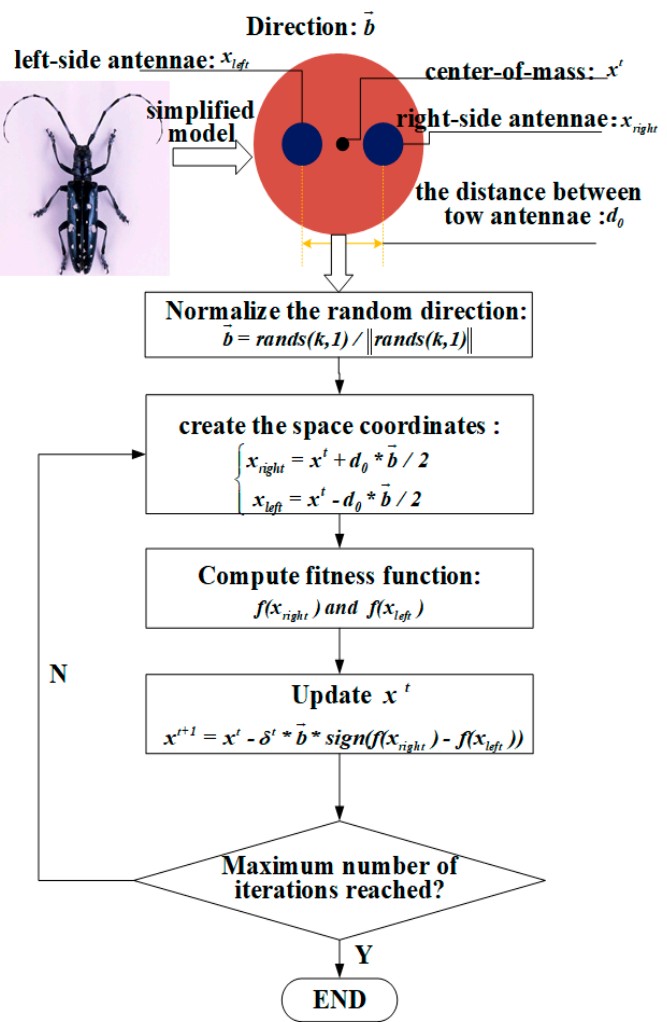

**Figure 3.** The flow chart of the beetle antennae search algorithm.

## 3. Results Establishment of the Simulation Model and Analysis of Fault Categories

This section may be divided by subheadings. It should provide a concise and precise description of the experimental results, their interpretation as well as the experimental conclusions that can be drawn. The simulation experiment is carried out for the two-level three-phase PWM rectifier, which converts 220 V AC voltage to 600 V DC voltage with a switching frequency of 10 kHz. Figure 4 shows the two-level three-phase PWM rectifier, which involves the main circuit and a control block diagram. The control block includes two current control loops and one DC-link voltage control loop. Furthermore, the AC-link current is converted to $d$ and $q$ axis current in a synchronous reference frame. The $q$-axis current is kept at zero to achieve unity power factor operating status. Additionally, the $d$-axis current is controlled to keep the DC-link voltage constant. The specifications of the circuit are listed in Table 1.

**Table 1.** Parameters Setting of Two-Level Three-Phase PWM Rectifier.

| Parameters | Value |
| --- | --- |
| Input AC voltage $U_{sa}$, $U_{sb}$, $U_{sc}$ | 220 V/50 Hz |
| Input boost inductance $L_s$ | 1 mH |
| Line resistance $R_s$ | 0.5 Ω |
| Switching frequency $f_s$ | 10 kHz |
| Injected current $i_{d,in}$ | 5 A |
| DC-link voltage $U$ | 600 V |
| DC-link capacitor $C$ | 4000 μF |
| DC-link resistance $R$ | 10 Ω |

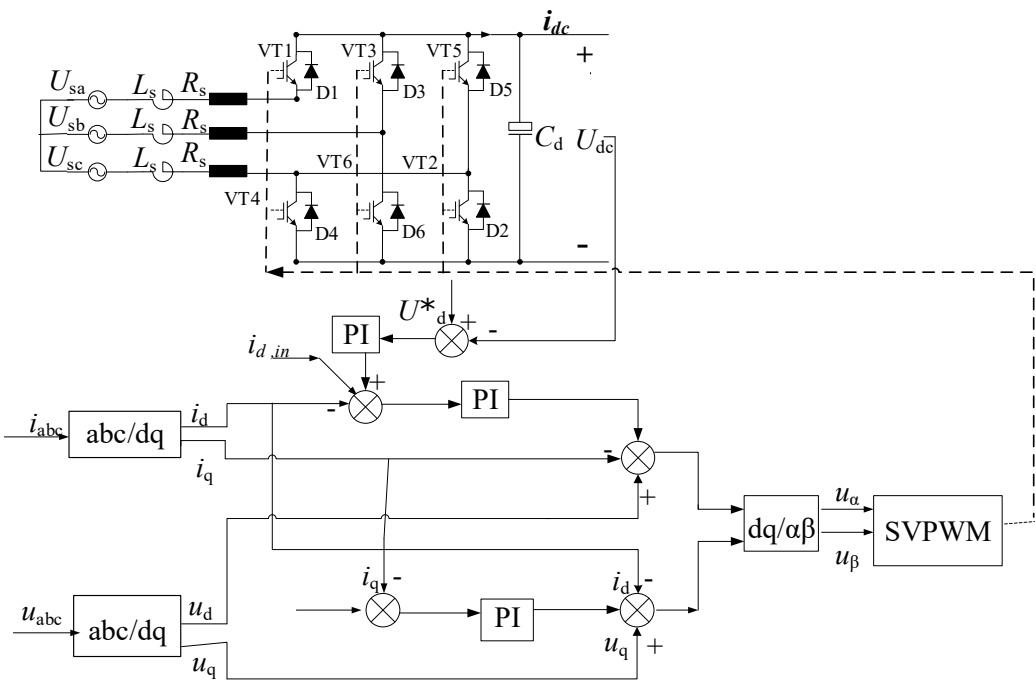

**Figure 4.** The simulation model of a three-phase PWM rectifier.

Since the multiple power semiconductor devices are unlikely to break down simultaneously, this paper only considers the fault of one power semiconductor device. According to the topology of the circuit, the circuit fault categories are divided into seven categories, including healthy condition and $VT_1$-$VT_6$ OCFs. Table 2 lists the fault modes, classification labels, and fault codes. More precisely, the classification label [0,1,0,0,0,0,0,0,0]T indicates that an OCF occurs at $VT_1$. In this paper, the MATLAB/Simulink model of the tested three-phase PWM rectifier is applied to the RT-LAB hardware-in-the-loop simulation system by PC, which reduces the difficulty of constructing the circuit and improves the reliability of the simulation system. Additionally, the data processing methods mentioned are implemented with MATLAB R2019a. As illustrated by Figure 5, the simulation experimental of the two-level three-phase PWM rectifier was built in the OP5600 simulator, which constructs a circuit response database containing multiple fault conditions and transmits the fault signal to the PC. The circuit response was captured at the output using a National Instruments (NI) USB-6212 data acquisition board. The data were recorded using LabVIEW on PC. The experiment operations and different fault settings are implemented in the OP5607 controller.

**Table 2.** Fault Modes and Classification Labels.

| Fault Modes | Classification Label | Fault Codes |
|---|---|---|
| Healthy condition | $[1,0,0,0,0,0,0]^T$ | 0 |
| $VT_1$ OCF | $[0,1,0,0,0,0,0]^T$ | 1 |
| $VT_2$ OCF | $[0,0,1,0,0,0,0]^T$ | 2 |
| $VT_3$ OCF | $[0,0,0,1,0,0,0]^T$ | 3 |
| $VT_4$ OCF | $[0,0,0,0,1,0,0]^T$ | 4 |
| $VT_5$ OCF | $[0,0,0,0,0,1,0]^T$ | 5 |
| $VT_6$ OCF | $[0,0,0,0,0,0,1]^T$ | 6 |

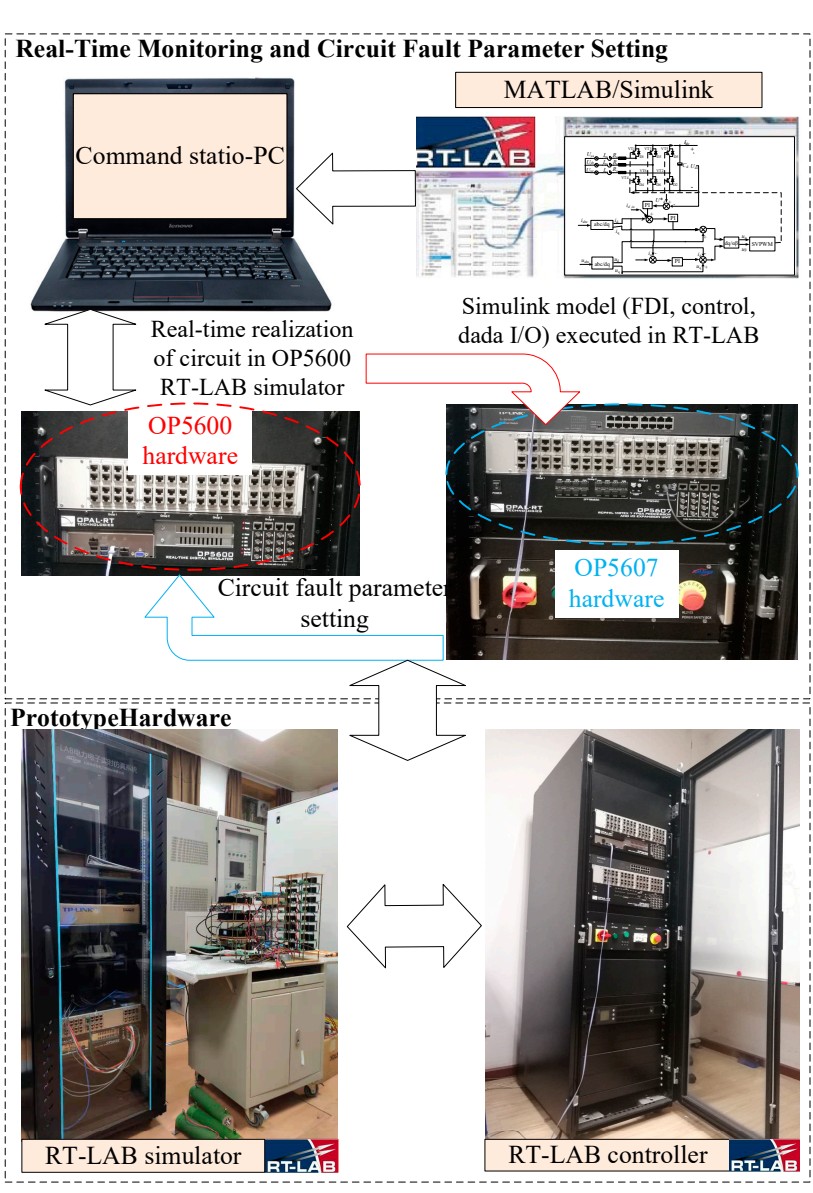

**Figure 5.** Real-time realization of the three-phase PWM rectifier with RT-LAB hardware-in-the-loop simulation.

## 4. Implementation Results and Discussion

### 4.1. Circuit Monitoring Signal Acquisition

Considering the operation of a two-level three-phase PWM rectifier under no-load condition, which means that the load side is virtually disconnected from DC-link. If the DC output voltage is

well controlled at its reference value, the d-axis current, which is a real power component, is zero and the DC-link voltage is kept constant, except for the switching frequency-related ripple components. However, since this is an equilibrium state, it is difficult to obtain any information on the system parameters for this condition. Thus, an AC signal at a specific frequency is injected into the *d*-axis current to solve the problem. Therefore, as shown in Figure 4, a suitable inject signal is used to excite the system for generating a fault signal, which is easy to extract and analyze. The injected current reference in the synchronous reference frame is given as follows:

$$i_{de,in} = 5\sin(50\pi t) \tag{22}$$

where $i_{dc\_in}$ is well controlled and is inversely transformed into the stationary reference frame, the AC current of the pulsed waveform with a fundamental frequency of 25 Hz flows through the line and the DC-link capacitors. Additionally, because the DC-link capacitor can compensate the DC-link output voltage harmonic changes and it may affect the accuracy of fault diagnosis, the DC-link output current is selected as the fault signal.

### 4.2. Fault Feature Extraction and Dimension Reduction

#### 4.2.1. Fault Feature Extraction Based on MEEMD

According to the previous fault modes setting of the circuit, the sampling time and the sampling frequency are set to 0.1 s and 100 kHz. Subsequently, the monitored current signal is decomposed into 7 IMF components via the MEEMD algorithm. Because the trend of IMF components with orders higher than seven tends to be flat and almost unchanged and contains little fault information. Therefore, the IMF1-IMF7 components are collected as fault features in this paper. As shown in Figure 6, the waveform of each IMF is different between different fault categories. The monitored current signal under healthy condition is closed to the fundamental frequency of 50 Hz, while the current waveform is distorted under the $VT_1$ OCF. To reduce the interference of irrelevant factors, the number of each fault category sample is set to 100, a total of $7 \times 100$ signal samples are obtained. Hereafter, the 17 features of each IMF component are calculated, which are represented in Table 3. Ultimately, the initial fault dataset **A** ($119 \times 700$) can be obtained.

**Table 3.** Fault Features and Computational Formula.

| No. | Fault Feature | Computational Formula | No. | Fault Feature | Computational Formula |
|-----|---------------|----------------------|-----|---------------|----------------------|
| F1 | Energy | $T_1 = \sum\limits_{i=1}^{n} |x(i)|^2$ | F10 | Impulse index | $T_{10} = \max x(i)/|T_3|$ |
| F2 | Complexity | $T_2 = Lempel\text{-}Ziv complexity$ | F11 | Peak index | $T_{11} = [\max x(i)]/T_4$ |
| F3 | Mean value | $T_3 = \sqrt{\frac{1}{n}\sum\limits_{i=1}^{n} x(i)}$ | F12 | Kurtosis index | $T_{12} = \left[1/n\sum\limits_{i=1}^{n} x(i)^4\right]/T_4^4$ |
| F4 | Root mean square value | $T_4 = \sqrt{\frac{1}{n}\sum\limits_{i=1}^{n} |x(i)|^2}$ | F13 | Frequency center | $T_{13} = \sum f\phi(f)/\sum f(f)$ |
| F5 | Standard deviation | $T_5 = \sqrt{\frac{1}{n}\sum\limits_{i=1}^{n} [x(i) - T_3]^2}$ | F14 | Mean square frequency | $T_{14} = \sum f^2\phi(f)/\sum f(f)$ |
| F6 | Skewness | $T_6 = \frac{1}{n-1}\sum\limits_{i=1}^{n} [x(i) - T_3]^3/T_5^3$ | F15 | Root mean square frequency | $T_{15} = \sqrt{T_{14}}$ |
| F7 | Kurtosis | $T_7 = \frac{1}{n-1}\sum\limits_{i=1}^{n} [x(i) - T_3]^4/\left(T_5^3 - 3\right)$ | F16 | Deviation frequency | $T_{16} = \sum (f - T_{13})^2\phi(f)/\sum \phi(f)$ |
| F8 | Coefficient of variation | $T_8 = T_4/|T_3|$ | F17 | Standard deviation frequency | $T_{17} = \sqrt{T_{16}}$ |
| F9 | Margin index | $T_9 = \max x(i)/\left[1/n\sum\limits_{i=1}^{n} \sqrt{|x(i)|}\right]^2$ | | | |

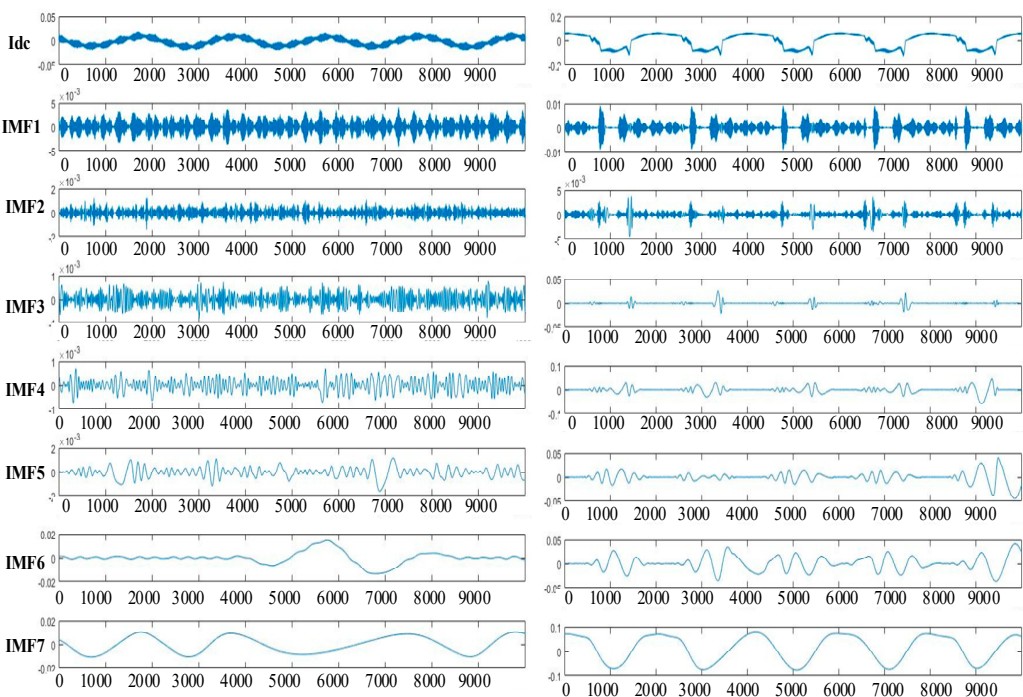

**Figure 6.** The output current and IMF components under normal and $VT_1$ OCF were obtained.

### 4.2.2. Dimensionality Reduction of Fault Feature Vectors Based on ERT and PCA

If all the features are imported into the classifier directly without further processing, it will increase the computational complexity. Hence, as shown in Figure 7, the ERT algorithm is used to calculate the VIM score of each feature to achieve feature selection from 119 kinds of fault features. Among them, the VIM values of 54 fault features are close to 0. The number of fault features for the IMF1 component, whose VIM values are equal to 0, is the least with a minimum of 3. It is proven that the IMF1 component contains the most fault feature information. On the contrary, the number of fault features for the IMF6 component, whose VIM values get close to 0, is the largest with a maximum of 14. It indicates that IMF6 contains the least fault feature information. Moreover, the 17 kinds of statistical parameters are also needed to be selected for retaining the excellent features. For instance, as for the mean and kurtosis index, the VIM values of each IMF component are not equal to 0, indicating that these two parameters are quite essential features to distinguish different fault categories. In contrast, as for the skewness and coefficient index, the VIM values for each IMF component are close to 0. For the Impulse index, although the VIM values calculated from IMF6 components are 0, the VIM value calculated from IMF2 is as high as 2.26, indicating that the appropriate fault features have a meaningful impact on the fault diagnosis results. Given the results, 119 kinds of fault features are ranked in descending order according to the value of VIM, and then the culling ratio is set as 0.6. Hence, the fault dataset **B** $(48 \times 700)$ with a dimension of 48 is obtained.

Additionally, this paper uses a t-distributed stochastic neighbor embedding (t-SNE) algorithm to map the high-dimensional data to a two-dimensional (2-D) space. The sample number of each fault category is 100. Due to the overlapping of points, the samples of the various fault categories shown in Figure 8a–d) appear to be different. The 2-D visualization of the initial fault feature dataset A is shown in Figure 8a. The label 0 represents the healthy condition; label 1 represents the $VT_1$ OCF, label 2 represents the $VT_2$ OCF. Additionally, in Figure 8a, there is no clear boundary between the samples of $VT_1$ OCF and $VT_3$ OCF. Figure 8b,c shows the t-SNE visualization of the dataset **B** obtained by PCA and ERT, respectively. Moreover, PC1 and PC2 represent the first and second principal components of the sample. The concentration of the fault in the same category in Figure 8b,c are lower than Figure 8a. The samples of $VT_1$ OCF and $VT_3$ OCF can be distinguished after feature selection via the PCA or ERT algorithm.

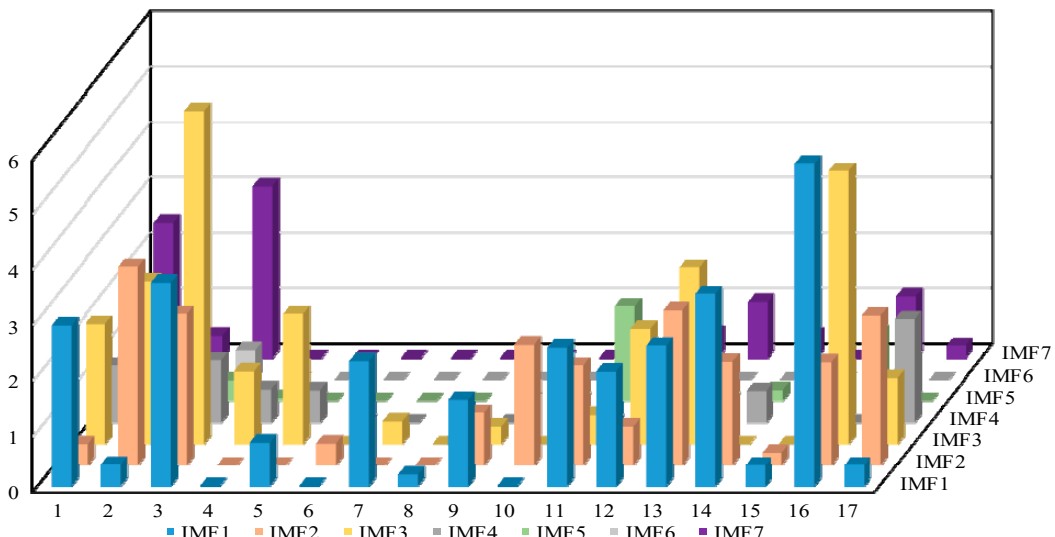

**Figure 7.** VIM score of 119 fault features.

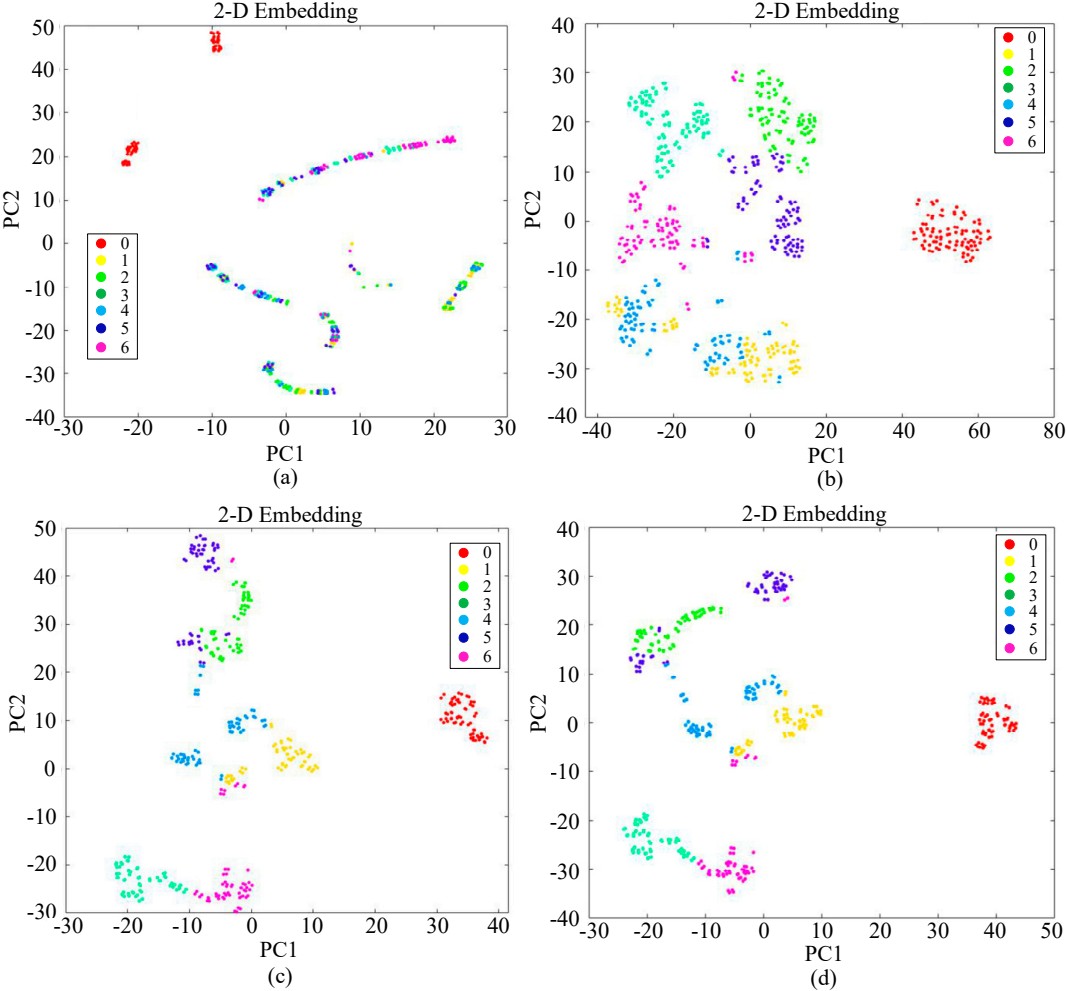

**Figure 8.** (**a**) t-SNE visualization of the dataset A; (**b**) t-SNE visualization of dataset B obtained by PCA; (**c**) t-SNE visualization of dataset C obtained by ERT; (**d**) t-SNE visualization of dataset D obtained by ERT + PCA.

Additionally, it can be seen from the comparison between Figure 8b and Figure 8c that the dimensionality reduction effect after ERT algorithm feature selection is significantly better than PCA.

The distance between different fault categories is increased, and the distance of samples in the same category is more concentrated. However, dimensionality reduction based on PCA is performed on the samples after ERT, as shown in Figure 8d, the dimensionality reduction effect has not been significantly improved. To further quantify the effects of different dimensionality reduction methods, the interclass distance and intraclass distance are calculated on the samples after different dimensionality reduction methods, as shown in Table 4.

**Table 4.** Dimensionality Reduction Effects of Different Methods.

| Dimensionality Reduction | Intraclass Distance | Interclass Distance | Feature Dimension |
|---|---|---|---|
| Initial fault feature | 38.268 | 15.687 | 96 |
| PCA | 37.544 | 15.654 | 35 |
| ERT | 4.799 | 6.484 | 32 |
| ERT+PCA | 4.550 | 6.441 | 25 |

*4.3. Fault Diagnosis Results of Different Classifiers*

Based on the above analysis, the dimension of each fault feature vector is 25. The constructed BAS-DBN classifier is used to accurately separate seven categories of faults and diagnose the fault category for the unknown samples. Additionally, a verification method, named K-fold cross-validation, is applied, which can randomly divide the samples into K repulsion subsets. Each K-1 subsets are randomly selected as a training set and the remaining one as a test set. Cross-validation repeated K times, and each subset is verified once. Finally, K times validation results are averaged to obtain the final accuracy. The advantage of K-fold cross-validation is that all the data will be applied as a training set and a test set, and the result will better reflect the model accuracy. K is set to 5, and a five-fold cross-validation method is applied in this paper. The fault diagnosis results of DBN is compared with shallow learning network to verify the performance of DBN. For BPNN, the parameters to be optimized are the initial connection weights and threshold values. As for the DBN, the number of hidden layer units is mainly optimized. Additionally, for the comparison of different optimization algorithms, GA is selected to compare with the BAS algorithm. The parameter settings are shown in Table 5.

**Table 5.** Parameters Setting of DBN.

| Network Structure Parameters | Parameters | Training Parameters | Parameters |
|---|---|---|---|
| No. of input layer units | 25 | Learning rate | 0.01 |
| No. of output layer units | 7 | Batch size | 150 |
| No. of hidden layers | 2 | Epoch | 100 |
| No. of hidden1 units | h1 (Optimization) | Activation | ReLU |
| No. of hidden2 units | h2 (Optimization) | Solver | L-BFGS |

4.3.1. GA-BPNN, BAS-BPNN and BAS-DBN

For the BPNN classifier, its structure is set to the most common three-layer architecture. The sigmoid function is selected as the transfer function of the hidden layer. After several trials, it is found that when the number of units in the hidden layer is set as 21, the fault diagnosis results perform better than others. Therefore, the structure of BPNN is set to 25-31-7. The number of parameters to be optimized is 385.

Dataset D based on ERT and PCA is divided into a training set and a testing set. The optimal initial network parameters are obtained by the optimization algorithm (GA or BAS). The fitness function is as follows:

$$err = \frac{1}{N}\sum_{i=1}^{N} \|y_{pre} - y_{true}\|^2 \tag{23}$$

where *err* is the error value, $y_{true}$ is the actual value, $y_{pre}$ is the predicted value, and *N* is the number of training samples.

The error evolution curves of GA-BPNN and BAS-BPNN are shown in Figure 9, the final training error obtained by the BAS optimization algorithm is 0.0655, while the training error obtained by the GA algorithm is 0.0458. Although the error of the former is higher than that of the latter, the number of iterations needed by the former is far less than that of the latter, and the optimal weights and thresholds can be obtained as soon as possible. It takes 106 iterations to seek the optimal parameters of BPNN by the GA algorithm. However, it takes 13 iterations to seek the optimal parameters of BPNN by the BAS algorithm. The number of input layer units in DBN is set to 25, and the number of output layer units is set to 7. The BAS algorithm optimizes the number of hidden1 and hidden2 layers units. The error evolution curve of BAS-DBN is shown in Figure 10, it can be known that when the number of units in the first hidden1 layer is 42, and the second hidden2 layer is 10, the value of error is 0.03301.

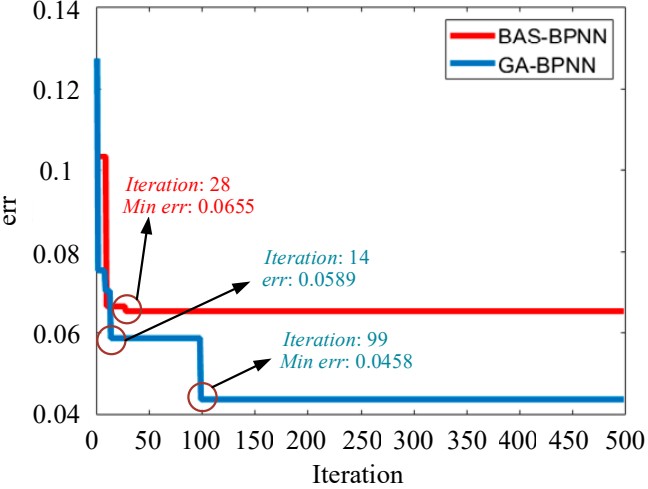

**Figure 9.** Error evolution curve of BAS-BPNN and GA-BPNN.

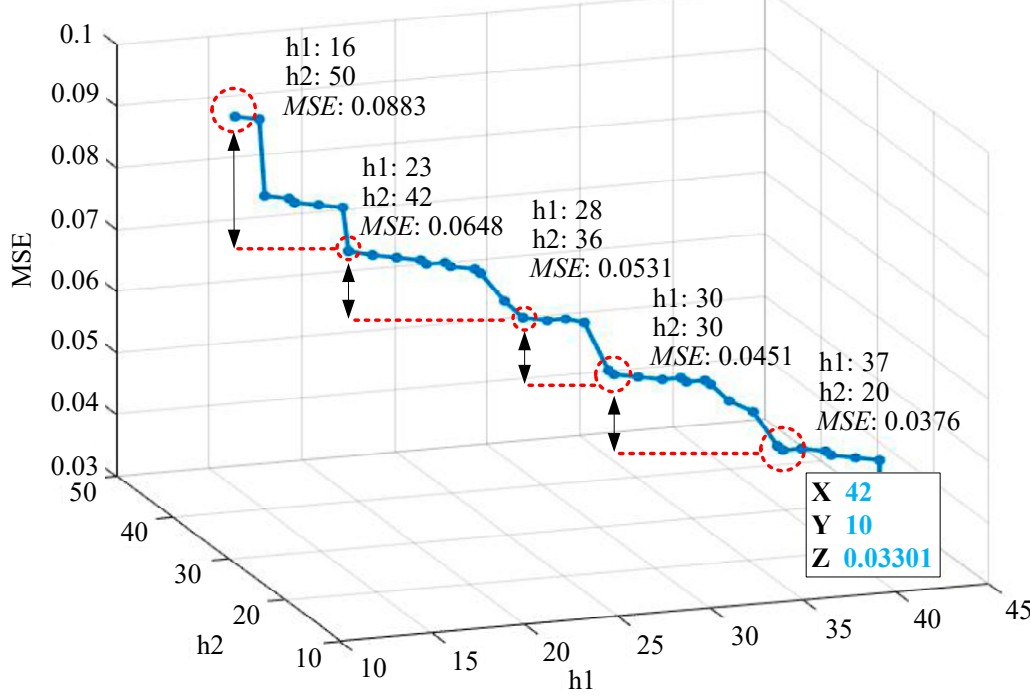

**Figure 10.** Error evolution curve of BAS-DBN.

### 4.3.2. Results of Fault Diagnosis

The fault diagnosis results of BPNN, GA-BPNN, BAS-BPNN, GA-DBN, and BAS-DBN are shown as Figure 11a–e. The points whose true label distributed between 0 and 1 represent the samples under healthy condition; the points between 1 and 2 represent the samples under $VT_1$ OCF, and so on. Similarly, the predicted label 0 represents the samples are under health condition, and the predicted label 1 represents the samples are under VT1 OCF. It can be known that BAS-DBN performs better than BPNN, GA-BPNN, BAS-BPNN, and GA-DBN. Among them, the BPNN classifier has a higher error rate in $VT_6$ and $VT_4$ OCF, and confusion appears between $VT_1$ OCF and $VT_3$ OCF. In the GA-BPNN classifier, the classification results of $VT_6$ and $VT_4$ OCF are improved. For GA-DBN, its performance is much better than BPNN, but it is a little worse than BAS-DBN. In the BAS-DBN classifier, not only the classification accuracy of $VT_2$, $VT_4$, and $VT_6$ OCF categories are guaranteed to reach 100%, but also the error rate of the other three fault categories reduced. The fault diagnosis accuracy of BAS-DBN is 98.43%, which is higher than other classifiers. The fault diagnosis accuracy of BPNN, GA-BPNN, and GA-DBN are 89.29%, 94%, and 97.28%, respectively.

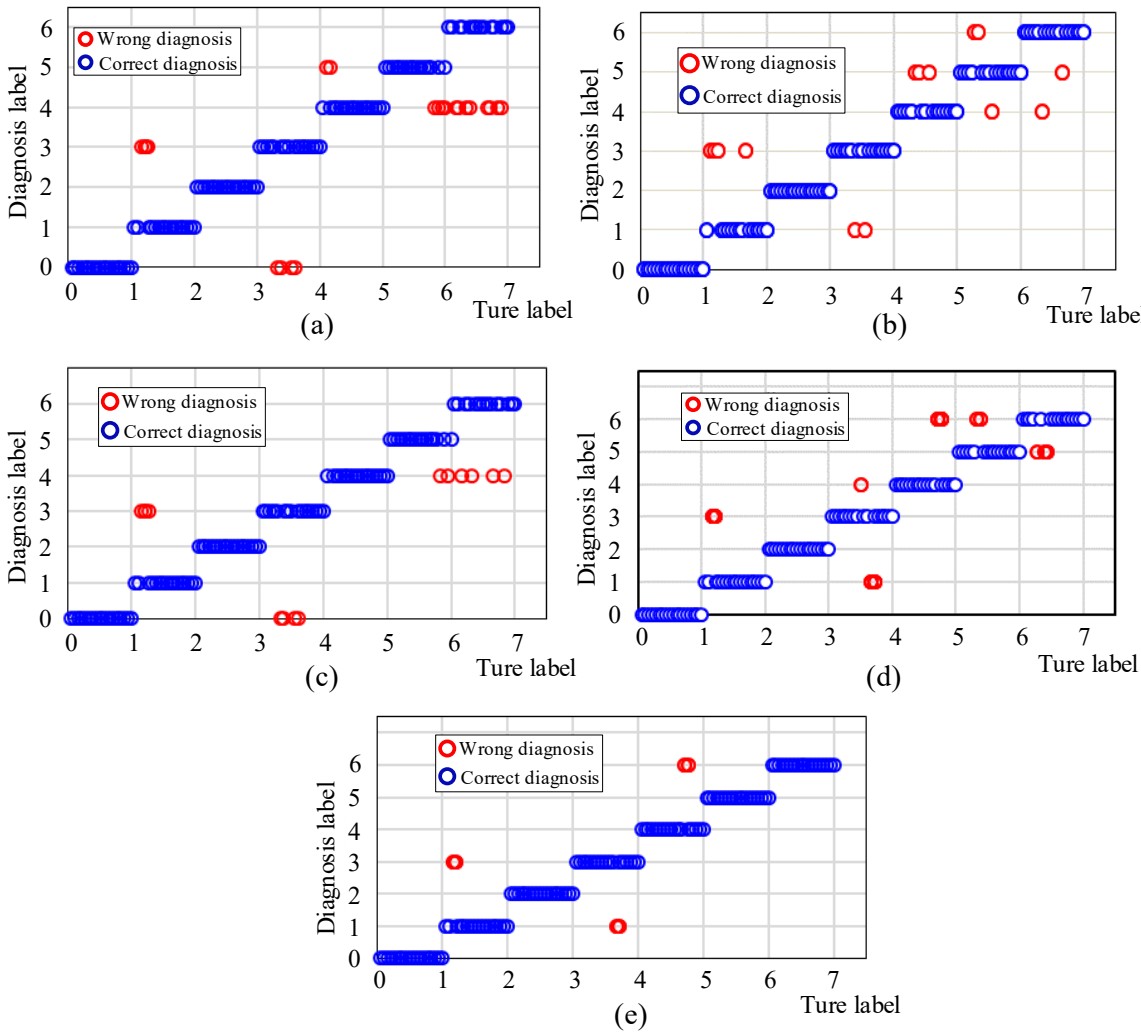

**Figure 11.** Fault diagnosis results of (**a**) BPNN; (**b**) GA-BPNN; (**c**) BAS-BPNN; (**d**) GA-DBN; (**e**) BAS-DBN.

As shown in Figure 12, to further verify the accuracy of each fault diagnosis method, we carried out 30 repeated experiments. It can be found that GA-BPNN is better than BAS-BPNN in diagnosing VT1, VT2, and VT3 OCFs, but BAS-BPNN performs better than GA-BPNN in diagnosing $VT_4$, $VT_5$,

and VT$_6$ OCFs. The BAS-DBN can efficiently diagnose every fault category, and its performance is relatively stable. The mean value of all fault diagnosis accuracy in every experiment is higher than 95%. The accuracy of fault diagnosis for BPNN, GA-BPNN, BAS-BPNN, GA-DBN, and BAS-DBN is 82.88%, 88.60%, 89.18%, 97.21%, and 98.58%, respectively.

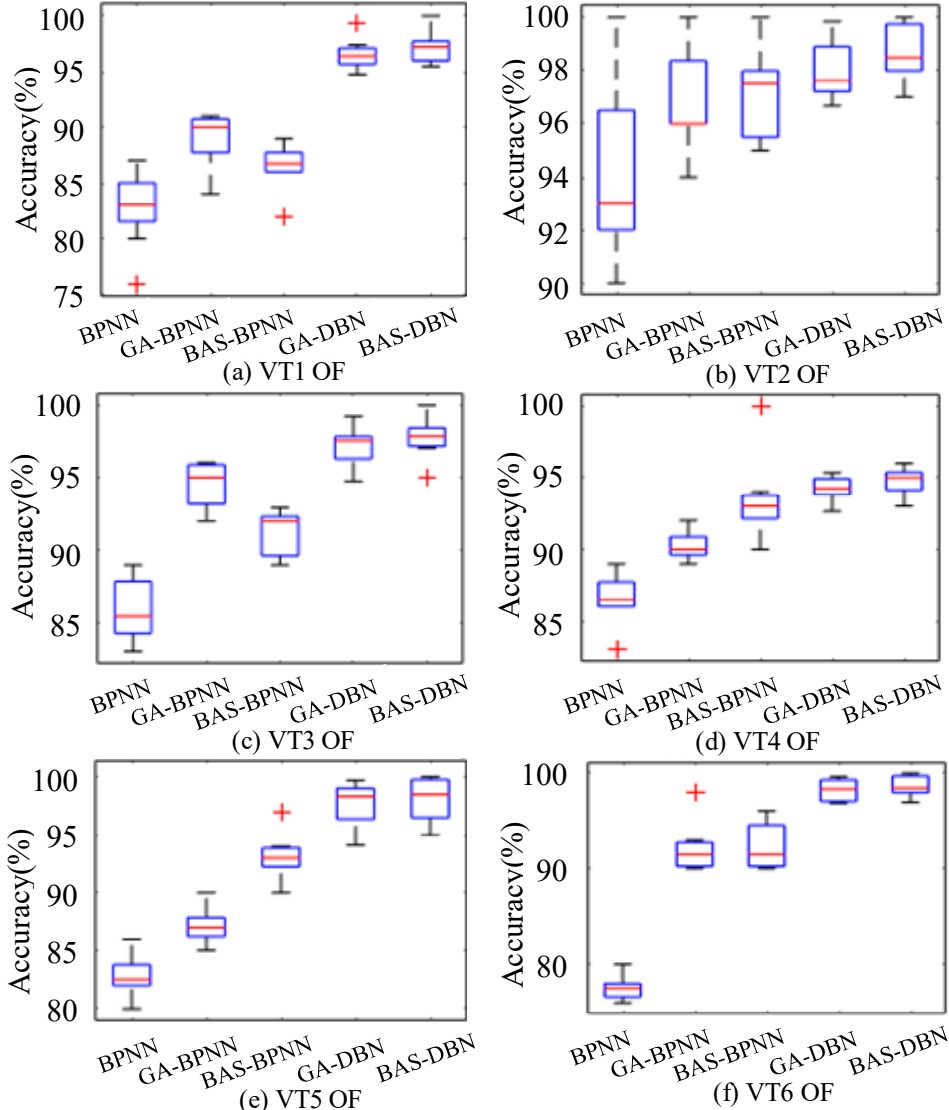

**Figure 12.** Fault diagnosis results of repeated tests.

## 5. Conclusions

In this paper, an OCFs diagnosis framework for a two-level, three-phase PWM rectifier is proposed by using MEEMD for feature extraction, ERT algorithm for the selection of the most relevant features, and BAS-DBN for fault diagnosis, which can reduce the fluctuation of the selected features as well as improve the accuracy of diagnosis. The effectiveness of the feature selection method is verified by measuring the intraclass and interclass distance between different samples. The features left behind are more conductive to fault diagnosis, and although the shallow neural network is used as the classifier, the fault diagnosis accuracy is higher than 90%. For the parameter setting of DBN, most papers choose the typical parameter setting or perform finite-time experiments to determine the number of hidden layer units. In our work, the optimization algorithm named BAS is used to train DBN, and the model which is most suitable for the converter fault recognition is obtained to ensure the highest accuracy of fault diagnosis.

**Author Contributions:** Conceptualization, methodology, software, validation, formal analysis, investigation, resources, B.D. and Y.H.; data curation, writing—original draft preparation, B.D.; writing—review and editing, B.D.; visualization, Y.Z. All authors have read and agreed to the published version of the manuscript.

**Funding:** This research was supported by the National Natural Science Foundation of China (Grant No. 51977153, 51977161, and 51577046), the State Key Program of National Natural Science Foundation of China (Grant No. 51637004), the national key research and development plan "important scientific instruments and equipment development" of China (Grant No. 2016YFF0102200), the Equipment research project in advance of China (Grant No. 41402040301).

**Conflicts of Interest:** The authors declare no conflict of interest.

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
