# Peer review of "Open-Circuit Fault Diagnosis of Three-Phase PWM Rectifier Using Beetle Antennae Search Algorithm Optimized Deep Belief Network"

_electronics, doi:10.3390/electronics9101570_

Round 1

Reviewer 1 Report

The authors propose an open-circuit faults diagnosis for a two-level three-phase pulse-width modulating rectifier by using MEEMD for feature extraction and extremely randomized trees algorithms in order to identify seven conditions.

  1. Language. The article is well written and clearly explained.
  2. Introduction. The introduction is well structured and contains relevant information.

  3. References. References are adequately discussed in a fair context. Most of them are updated. Key references on the field are cited. However, the references section could be improved. There are 19 references of 37 with more than five years old, so some of them could be updated.

  4. Structure. The manuscript is well structured, containing all relevant sections required in a technical paper. The manuscript study a real-time realization of the three-phase PWM rectifier with RT-LAB hardware-in-the-loop simulation, where the obtained results are examined and explained correctly.

  5. Results and discussion. Results are appropriately discussed, compared, and presented in the appropriate context to findings from other recently published methods.

  6. Novelty/contribution. The manuscript presents fair novelty and contributions to the field.

  7. I appreciate the hard work done by the authors and they propose an interesting methodology. Having said that, the following comments bears significance concerning this paper:

    a) In the simulation analysis, I do not find what kind of noise is used. Could the authors explain if some noise is considered? Two types of uncertainties should be considered in simulations; the first one is the measurement noise (it is Gaussian), and the second is a noise intrinsic to the system, from unidentified sources (it is also introduced within the simulations). The obtained results from these noise considerations lead to realistic observations both in terms of the performance of the experiment and in terms of testing the proposed methodology. The authors should explain and justify the choice of some noise.

    b) How much time does the proposed methodology takes to provide results?

    c) The study could identify seven conditions. My question is if your proposed methodology could identify more than seven conditions, or could it be possible to identify multiple combined faults?

    d) On page 11, you state, “It is necessary to assume that the two-level three-phase PWM rectifier is operating under no load conditions.” Does it mean that your methodology does not work under load conditions?

    e) How is chosen the sampling frequency in 10 kHz?

    f) Add the labels and units in Fig. 8. Other figures do not show the units.

Author Response

Response to Reviewer 1 Comments

Dear reviewer and editor:

We gratefully thank you for the precious time you spent making constructive remarks and useful suggestions, which have significantly raised the quality of the manuscript and have enabled us to improve the manuscript. Each suggested revision and comment, brought by you was considered. All revisions in this manuscript are clearly highlighted by the "Track Changes" function in Microsoft Word. Below the comments are response point by point, and the revisions are indicated by line numbers of the revised manuscript.

Point 1: The authors propose an open-circuit faults diagnosis for a two-level three-phase pulse-width modulating rectifier by using MEEMD for feature extraction and extremely randomized trees algorithms in order to identify seven conditions.

Language. The article is well written and clearly explained.

Introduction. The introduction is well structured and contains relevant information.

References. References are adequately discussed in a fair context. Most of them are updated. Key references on the field are cited. However, the references section could be improved. There are 19 references of 37 with more than five years old, so some of them could be updated.

Structure. The manuscript is well structured, containing all relevant sections required in a technical paper. The manuscript study a real-time realization of the three-phase PWM rectifier with RT-LAB hardware-in-the-loop simulation, where the obtained results are examined and explained correctly.

Results and discussion. Results are appropriately discussed, compared, and presented in the appropriate context to findings from other recently published methods.

Novelty/contribution. The manuscript presents fair novelty and contributions to the field.

Response 1: Thanks. Thank you for your approval of this paper.

Yes, in “Section”, the state-of-the-art feature extraction, feature selection, and fault diagnosis. approaches are reviewed. In addition, considering the Reviewer’s suggestion, we have added excellent references for fault diagnosis in the recent years. The updated references are as follows:

References

[7] Yaghoubi M, Moghani J S, Noroozi N, et al. "IGBT Open-Circuit Fault Diagnosis in a Quasi-Z-Source Inverter," IEEE Transactions on Industrial Electronics, 2018, 66, 2847-2856. doi: 10.1109/TIE.2018.2847709.

[11] Xia Y, Xu Y, Gou B. "A Data-Driven Method for IGBT Open-Circuit Fault Diagnosis Based on Hybrid Ensemble Learning and Sliding-Window Classification," IEEE Transactions on Industrial Informatics, 2020, 16, 5223-5233, doi: 10.1109/TII.2019.2949344.

[13] Kumar G K, Elangovan D. "Review on fault-diagnosis and fault-tolerance for DC–DC converters," IET Power Electronics, 2020, 13, 1-13, doi: 10.1049/iet-pel.2019.0672.

[22] Nie X, Liu S, Xie G, "A Novel Autoencoder with Dynamic Feature Enhanced Factor for Fault Diagnosis of Wind Turbine," Electronics, 2020, 74, doi:10.3390/electronics9040600.

[26] Burriel-Valencia J, Puche-Panadero R, Martinez-Roman J, et al. "Automatic Fault Diagnostic System for Induction Motors under Transient Regime Optimized with Expert Systems," Electronics, 2018, 8, doi: 10.3390/electronics8010006.

[31] Liu Y, Chai Y, Wei S, et al. “Circuit Fault Diagnosis Method of Wind Power Converter with Wavelet-DBN,” Chinese Intelligent Systems Conference, 2017, 460, 623-633, doi: 10.1007/978-981-10-6499-9-60.

[34] Acosta M R C , Ahmed S , Garcia C E , et al. "Extremely Randomized Trees-Based Scheme for Stealthy Cyber-Attack Detection in Smart Grid Networks," IEEE Access, 2020, 8:19921-19933, doi: 10.1109/ACCESS.2020.2968934.

Point 2: I appreciate the hard work done by the authors and they propose an interesting methodology. Having said that, the following comments bears significance concerning this paper:

In the simulation analysis, I do not find what kind of noise is used. Could the authors explain if some noise is considered? Two types of uncertainties should be considered in simulations; the first one is the measurement noise (it is Gaussian), and the second is a noise intrinsic to the system, from unidentified sources (it is also introduced within the simulations). The obtained results from these noise considerations lead to realistic observations both in terms of the performance of the experiment and in terms of testing the proposed methodology. The authors should explain and justify the choice of some noise.

Response 2: Thanks. Thanks for your comments.

In this paper, we have considered the influence of noise on the measured signal of the circuit system. We use a noise inject to the system, from unidentified sources (it is also introduced within the simulations). A specific signal needs to be injected to the circuit system, for example, a regulated AC current with a low frequency can be injected into the input side of the AC/DC PWM converter. In Fig.4 ,the injected current reference in the synchronous reference frame is given as:

where  is well controlled and is inversely transformed into the stationary reference frame, the AC current of the pulsed waveform with a fundamental frequency of 25Hz flows through the line and the DC-link capacitors. Additionally, because the DC-link capacitor can compensate the DC-link output voltage harmonic changes and it may affect the accuracy of fault diagnosis, the DC-link output current is selected as the fault signal.

Figure 4. The simulation model of a three-phase PWM rectifier.

Point 3: How much time does the proposed methodology takes to provide results?

Response 3: Thanks. Thanks for your comments. The time for data acquisition, feature extraction and feature selection may be longer, and it must take time to train the DBN network model. But for a trained and optimized DBN network model, it takes only a few seconds to get the diagnosis results by putting the selected features into the network for testing. The related researches using intelligent algorithm and network model to diagnose different fault can refer to the following papers, for the trained model, the diagnosis results are only a few seconds.

References

[1] W He; YG He ; B Li, et al. "A Naive-Bayes-based fault diagnosis approach for analog circuit by using image-oriented feature extraction and selection technique," IEEE Access, 2020, 8, 5065-5079. doi: 10.1109/ACCESS.2018.2888950.

[2] Sun Q, Wang Y , Jiang Y. "A Novel Fault Diagnostic Approach for DC-DC Converters Based on CSA-DBN," IEEE Access, 2017,1, doi: 10.1109/ACCESS.2017.2786458.

[3] Wang T, Xu H, Han J, et al. "Cascaded H-Bridge Multilevel Inverter System Fault Diagnosis Using a PCA and Multiclass Relevance Vector Machine Approach," IEEE Transactions on Power Electronics, 2015, 30, 7006-7018, doi: 10.1109/TPEL.2015.2393373.

[4] Binu D, Kariyappa B S. "RideNN: A New Rider Optimization Algorithm-Based Neural Network for Fault Diagnosis in Analog Circuits," IEEE Transactions on Instrumentation and Measurement, 2019, 68, 2-26.

[5] Tiancheng S, Yigang H, Fangming D, et al. "Online diagnostic method of open-switch faults in PWM voltage source rectifier based on instantaneous AC current distortion," IET Electric Power Applications, 2017, 12, 447-454, DOI: 10.1049/iet-epa.2017.0438.

Point 4: The study could identify seven conditions. My question is if your proposed methodology could identify more than seven conditions, or could it be possible to identify multiple combined faults?

Response 4: Thanks. Thanks for your comments. Because the ability of power electronic devices to withstand overvoltage and overcurrent is much weaker than other circuits, a very short period of time overvoltage and overcurrent will lead to permanent damage to the device and instant failure of power electronic system. Therefore, this paper chooses IGBT as the research object, whose failure rate is only lower than the electrolytic capacitance. Since the probability of two switch devices to fail at the same time is small, this paper only studies the case of single component failure. The proposed methodology can diagnose multiple combined faults. In principle, no matter multiple fault or single fault, as long as the output voltage waveform of different fault condition is different, after feature extraction and feature selection, combined with the fault diagnosis method proposed in this paper, the multiple fault can be diagnosis can be diagnosed.

Point 5: On page 11, you state, “It is necessary to assume that the two-level three-phase PWM rectifier is operating under no load conditions.” Does it mean that your methodology does not work under load conditions?

Response 5: Thanks. Thanks for your comments. The proposed diagnosis method also works under load conditions. Maybe the words we wrote is ambiguous. So, in the revised manuscript, we have already deleted it. What the authors want to express in the paper is as follows:

Assuming that the output power is zero under no load condition, if the output DC voltage is a constant value, the d-axis current should be controlled to zero, and the whole circuit system is in a balanced state, which is not conducive to fault signal extraction. Therefore, in order to collect effective fault feature at DC-link, an AC current of specific frequency is injected into d-axis component of AC-link current of PWM rectifier, and ripple voltage of the same frequency will be generated at DC-link. Due to the existence of DC output capacitance, the voltage drops and harmonic changes caused by some faults can be compensated, thus affecting the normal fault detection. Therefore, the DC side current signal is selected as the fault characteristic signal [15].

Point 6: How is chosen the sampling frequency in 10 kHz?

Response 6: Thanks. We highly appreciate the constructive comments and suggestions of reviewers. Sincerely thanks again for Reviewers’ warm work earnestly. The simulation experiment is carried out for the two-level three-phase PWM rectifier, which converts 220V AC voltage to 600V DC voltage with a switching frequency of 10 kHz. We made a mistake. The sampling frequency should be 100kHz instead of 10kHz. The sampling time is set to 0.1s. Therefore, the number of samples is 100*103*0.1=10000. As shown in Fig 6, there are 10000 sampling points in current waveform and IMF components.

Figure 6. The output current and IMF components under normal and VT1 OCF were obtained.

Point 7: Add the labels and units in Fig. 8. Other figures do not show the units.

Response 7: Thanks. Thanks for your comments. In the revised manuscript, the low-resolution image and the figure text sizes are improved. According to the problem the Reviewer pointed out, the labels and units in Fig. 8 are Added. A detail definition of the symbol is added to Fig. 8, t-SNE diagrams actually decompose the data using PCA, and then show the first and the second principal components (PC1 and PC2).

For example, the amendment is made as follows:

Figure 8. (a) t-SNE visualization of the dataset A; (b) t-SNE visualization of dataset B obtained by PCA; (c) t-SNE visualization of dataset C obtained by ERT; (d) t-SNE visualization of dataset D obtained by ERT+PCA.

-----------------------------------------end of comments------------------------------------------

Special thanks to you for your good comments. We tried our best to improve the manuscript. We appreciate for Reviewers’ warm work earnestly and hope that the correction will meet with approval. Once again, thank you very much for your comments and suggestions.

Best regards!

Yours Sincerely Authors

16thSeptember, 2020

Reviewer 2 Report

In this article authors present modern fault diagnosis strategy basing on modified ensemble empirical mode decomposition (MEEMD) and Beetle Antennae Search (BAS) algorithm optimized Deep Belief Network (DBN).Described methodology and theoretical algorithms of feature extraction, feature selection, and fault diagnosis.
There is presented model of a two-level three-phase PWM rectifier and the fault categories.
Experiments results of various methods of classifications compared with BAS-DBN are shown. Authors clearly described conducted researches. Methods which are applied are appropriate. Research results are described successfully. Bibliography is very well documented and contains a lot of papers which have been published since 2018. Despite of huge number of pictures there are well positioned inside the paper and are readable. There are minor editorial issues, e.g. verse 149 – missing colon sign.

Author Response

Response to Reviewer 2 Comments

Dear reviewer and editor:

We gratefully thank you for the precious time you spent making constructive remarks and useful suggestions, which have significantly raised the quality of the manuscript and have enabled us to improve the manuscript. Each suggested revision and comment, brought by you was considered. All revisions in this manuscript are clearly highlighted by the "Track Changes" function in Microsoft Word. Below the comments are response point by point, and the revisions are indicated by line numbers of the revised manuscript.

Point 1: In this article authors present modern fault diagnosis strategy basing on modified ensemble empirical mode decomposition (MEEMD) and Beetle Antennae Search (BAS) algorithm optimized Deep Belief Network (DBN). Described methodology and theoretical algorithms of feature extraction, feature selection, and fault diagnosis.

There is presented model of a two-level three-phase PWM rectifier and the fault categories.

Experiments results of various methods of classifications compared with BAS-DBN are shown. Authors clearly described conducted researches. Methods which are applied are appropriate. Research results are described successfully. Bibliography is very well documented and contains a lot of papers which have been published since 2018. Despite of huge number of pictures there are well positioned inside the paper and are readable. There are minor editorial issues, e.g. verse 149 – missing colon sign.

Response 1: Thanks. Thanks for your comments.

  1. In the original manuscript, there are minor editorial issues, and the amendments are made as follows:

(1) Page 1, Abstract, line 18, add "the".(2) Page 1, Introduction, line 39, replace "changes" by "to change".(3) Page 1, Introduction, line 69, replace "were" by "was".(4) Page 1, Introduction, line 70-80, add “another powerful signal processing method for non-linear and non-stationary signals named Empirical Mode Decomposition (EMD), ensemble empirical mode decomposition (EEMD) [18] and complete ensemble empirical mode decomposition (CEEMD) [19], have been widely used to solve fault diagnosis of rotating machinery and circuit system. Additionally, compared with wavelet transform where the basic functions are fixed, EMD-based method decomposes signals according to time-scale characteristics of data without setting any basis function in advance, which has stronger local stationary. However, the EEMD and CEEMD algorithms are time-consuming, the number of iterations has a great impact on the decomposition effect. Therefore, this paper uses modified ensemble empirical mode decomposition (MEEMD) [20, 21] algorithm to extract fault feature of the three-phase PWM rectifier, which not only suppress the mode confusion in decomposition process, but also reduce the calculation amount.”(5) Page 3, Introduction, line 100, replace "ANN" by "artificial neural network (ANN)".(6) Page 3, Introduction, line 127, add “As an improved EMD-based algorithm, MEEMD overcomes the shortcomings of EEMD and CEEMD. It has less computation time and higher reconstruction accuracy when decomposing the original signal into more representative intrinsic mode function (IMF) components.”(7) Page 3, Introduction, line 142, replace "experiment" by "experimental".(8) Page 4, Section 2, line 146, add a colon sign in verse 149.(9) Page 4, Section 2, line 161-162, replace "BAS-DBN" by "beetle antennae search algorithm optimized deep belief network".

  • Page 5, Section 2.1, line 181, add " Otherwise".
  • Page 5, Section 2.1, line 182, replace "an abnormal component experiment" by " abnormal".
  • Page 5, Section 2.1, line 184, add “Finally”.

(13) Page 7, Section 2.3, line 248, replace "with respect to" by "concerning".(14) Page 11, Section 4.1, line 341, replace "In order to" by "To".(15) Page 13, Section 4.2.2, line 372, replace "shows" by "show".(16) Page 13, Section 4.2.2, line 375, replace "is" by "are".

  1. In addition, we have added the references [7], [11], [13], [22], [26], [31], [34] for fault diagnosis in the recent years. The updated references are as follows:

References

[7] Yaghoubi M, Moghani J S, Noroozi N, et al. "IGBT Open-Circuit Fault Diagnosis in a Quasi-Z-Source Inverter," IEEE Transactions on Industrial Electronics, 2018, 66, 2847-2856. doi: 10.1109/TIE.2018.2847709.

[11] Xia Y, Xu Y, Gou B. "A Data-Driven Method for IGBT Open-Circuit Fault Diagnosis Based on Hybrid Ensemble Learning and Sliding-Window Classification," IEEE Transactions on Industrial Informatics, 2020, 16, 5223-5233, doi: 10.1109/TII.2019.2949344.

[13] Kumar G K, Elangovan D. "Review on fault-diagnosis and fault-tolerance for DC–DC converters," IET Power Electronics, 2020, 13, 1-13, doi: 10.1049/iet-pel.2019.0672.

[22] Nie X, Liu S, Xie G, "A Novel Autoencoder with Dynamic Feature Enhanced Factor for Fault Diagnosis of Wind Turbine," Electronics, 2020, 74, doi:10.3390/electronics9040600.

[26] Burriel-Valencia J, Puche-Panadero R, Martinez-Roman J, et al. "Automatic Fault Diagnostic System for Induction Motors under Transient Regime Optimized with Expert Systems," Electronics, 2018, 8, doi: 10.3390/electronics8010006.

[31] Liu Y, Chai Y, Wei S, et al. “Circuit Fault Diagnosis Method of Wind Power Converter with Wavelet-DBN,” Chinese Intelligent Systems Conference, 2017, 460, 623-633, doi: 10.1007/978-981-10-6499-9-60.

[34] Acosta M R C , Ahmed S , Garcia C E , et al. "Extremely Randomized Trees-Based Scheme for Stealthy Cyber-Attack Detection in Smart Grid Networks," IEEE Access, 2020, 8:19921-19933, doi: 10.1109/ACCESS.2020.2968934.

  1. In the revised manuscript, low resolution images are improved, and the figure text sizes are increased.

For example, the amendments (Figs. 8-12) are made as follows:

Figure 8. (a) t-SNE visualization of the dataset A; (b) t-SNE visualization of dataset B obtained by PCA; (c) t-SNE visualization of dataset C obtained by ERT; (d) t-SNE visualization of dataset D obtained by ERT+PCA.

Figure 9. Error evolution curve of BAS-BPNN and GA-BPNN.

Figure 10. Error evolution curve of BAS-DBN.

Figure 11. Fault diagnosis results of (a) BPNN; (b) GA-BPNN; (c) BAS-BPNN; (d) GA-DBN; (e) BAS-DBN.

Figure 12. Fault diagnosis results of repeated tests.

-----------------------------------------end of comments------------------------------------------

Finally, we highly appreciate the constructive comments and suggestions of reviewers. Sincerely thanks again for Reviewers’ warm work earnestly.

Best regards!

Yours Sincerely Authors

16thSeptember, 2020

Reviewer 3 Report

The manuscript proposes the fault diagnosis of three-phase pulse-width modulating rectifier via beetle antennae search algorithm optimized deep belief network. The contents of the manuscript are interesting and most probably of interest to the readers. The article is also timely with the increasing trend in solutions offered by machine learning methods. 

Generally speaking, in scientific literature I would always prefer high quality images in the manuscripts, e.g. vector graphics or high resolution images. In the current version of the manuscript most of the figures suffer from low resolution. I would definitely update the figures to higher resolution prior to publication. Please make clear figures or do not place them at all in the manuscript. Unclear figures or equations make the paper seem like it was rushed and decreases the readability of the paper. 

For example, in Figure 8 (similarly figures 9-12) the clusters are almost indistinguishable. These figures must be either improved or removed. Also, increase the figure text sizes, very small font size makes difficult to interpret the figures. 

Author Response

Response to Reviewer 3 Comments

Dear reviewer and editor:

We gratefully thank you for the precious time you spent making constructive remarks and useful suggestions, which have significantly raised the quality of the manuscript and have enabled us to improve the manuscript. Each suggested revision and comment, brought by you was considered. All revisions in this manuscript are clearly highlighted by the "Track Changes" function in Microsoft Word. Below the comments are response point by point, and the revisions are indicated by line numbers of the revised manuscript.

Point 1: The manuscript proposes the fault diagnosis of three-phase pulse-width modulating rectifier via beetle antennae search algorithm optimized deep belief network. The contents of the manuscript are interesting and most probably of interest to the readers. The article is also timely with the increasing trend in solutions offered by machine learning methods.

Generally speaking, in scientific literature I would always prefer high quality images in the manuscripts, e.g. vector graphics or high resolution images. In the current version of the manuscript most of the figures suffer from low resolution. I would definitely update the figures to higher resolution prior to publication. Please make clear figures or do not place them at all in the manuscript. Unclear figures or equations make the paper seem like it was rushed and decreases the readability of the paper.

For example, in Figure 8 (similarly figures 9-12) the clusters are almost indistinguishable. These figures must be either improved or removed. Also, increase the figure text sizes, very small font size makes difficult to interpret the figures.

Response 1: Thanks. Thanks for your comments. In the revised manuscript, low resolution images are improved, and the figure text sizes are increased.

For example, the amendments (Figs. 8-12) are made as follows:

Figure 9. Error evolution curve of BAS-BPNN and GA-BPNN.

Figure 8. (a) t-SNE visualization of the dataset A; (b) t-SNE visualization of dataset B obtained by PCA; (c) t-SNE visualization of dataset C obtained by ERT; (d) t-SNE visualization of dataset D obtained by ERT+PCA.

Figure 10. Error evolution curve of BAS-DBN.

Figure 11. Fault diagnosis results of (a) BPNN; (b) GA-BPNN; (c) BAS-BPNN; (d) GA-DBN; (e) BAS-DBN.

Figure 12. Fault diagnosis results of repeated tests.

--------------------------------------------end of comments------------------------------------------------

Finally, we highly appreciate the constructive comments and suggestions of reviewers. Sincerely thanks again for Reviewers’ warm work earnestly.

Best regards!

Yours Sincerely Authors

16thSeptember, 2020

Reviewer 4 Report

The paper is concerned with an open-circuit fault diagnosis approach applied to a two-level three-phase PWM rectifier, which is based on: a modified ensemble empirical mode decomposition for feature extraction; an Extremely Randomized Trees algorithm for selection of the most relevant features; and a Beetle Antennae Search algorithm to optimized Deep Belief Networks, aiming the fluctuation of the selected features and the improvement of the diagnosis accuracy. A suitable survey of published material related with the paper’s topic is included. The paper is very well written and structured, being easy to follow the authors’ ideas and makes a good balance between the description of the theoretical stuff needed to understand the authors’ proposed approach and the discussion about the results achieved considering a case study used to evaluate its robustness. Thus, in the reviewer opinion the paper is recommendable for publication as it is.

Author Response

Response to Reviewer 4 Comments

Dear reviewer and editor:

We gratefully thank you for the precious time you spent making constructive remarks and useful suggestions. Each suggested revision and comment, brought by you was considered. All revisions in this manuscript are clearly highlighted by the "Track Changes" function in Microsoft Word. Below the comments are response point by point, and the revisions are indicated by line numbers of the revised manuscript.

Point 1: The paper is concerned with an open-circuit fault diagnosis approach applied to a two-level three-phase PWM rectifier, which is based on: a modified ensemble empirical mode decomposition for feature extraction; an Extremely Randomized Trees algorithm for selection of the most relevant features; and a Beetle Antennae Search algorithm to optimized Deep Belief Networks, aiming the fluctuation of the selected features and the improvement of the diagnosis accuracy. A suitable survey of published material related with the paper’s topic is included. The paper is very well written and structured, being easy to follow the authors’ ideas and makes a good balance between the description of the theoretical stuff needed to understand the authors’ proposed approach and the discussion about the results achieved considering a case study used to evaluate its robustness. Thus, in the reviewer opinion the paper is recommendable for publication as it is.

Response 1: Thanks. Thank you for your approval of this paper!

-------------------------------------------------end of comments-------------------------------------------

Finally, we highly appreciate the constructive comments and suggestions of reviewers. Sincerely thanks again for Reviewers’ warm work earnestly.

Best regards!

Yours Sincerely Authors

16thSeptember, 2020
